# Helping Rabbits Cope with Veterinary Acts and Vaccine-Related Stress: The Effects of the Rabbit Appeasing Pheromone (RAP)

**DOI:** 10.3390/ani14233549

**Published:** 2024-12-09

**Authors:** Pietro Asproni, Elisa Codecasa, Miriam Marcet-Rius, Juliane Demellier, Estelle Descout, Marine Verbaere, Orane Vinck, Patrick Pageat, Alessandro Cozzi

**Affiliations:** 1Tissue Biology and Chemical Communication Department, IRSEA, Institute of Research in Semiochemistry and Applied Ethology, 84400 Apt, France; m.verbaere@irsea-institute.com; 2Ethics, Legislation & Animal Welfare Department, IRSEA, Institute of Research in Semiochemistry and Applied Ethology, 84400 Apt, France; e.codecasa@irsea-institute.com (E.C.); m.marcet@irsea-institute.com (M.M.-R.); j.demellier@irsea-institute.com (J.D.); 3Statistics and Data Management Service, IRSEA, Institute of Research in Semiochemistry and Applied Ethology, 84400 Apt, France; e.descout@irsea-institute.com; 4Vertebrate Animals Experimentation Service, IRSEA, Institute of Research in Semiochemistry and Applied Ethology, 84400 Apt, France; o.francois@outlook.com; 5Research and Education Board, IRSEA, Institute of Research in Semiochemistry and Applied Ethology, 84400 Apt, France; p.pageat@group-irsea.com (P.P.); a.cozzi@irsea-institute.com (A.C.)

**Keywords:** adaptation, behaviour, pets, pheromones, rabbit appeasing pheromone, stress, vaccine-related stress, welfare

## Abstract

Pet rabbits are exposed to various stressful situations from the first days of life. Rabbits need to adapt to all these particular circumstances, an important challenge with which they need to be helped. In particular, veterinary procedures are known to be an important source of stress. Previous studies on various species showed that maternal appeasing pheromones can help animals to better adapt to stressful situations. As in other species, the analogue of this pheromone in rabbits, rabbit appeasing pheromone (RAP), is known and used. In our study, we wanted to evaluate whether continuous exposure to RAP could facilitate the adaptation process in order to better face veterinary consultations and vaccinations using some behavioural indicators and analysing videos recorded during vaccination. Statistical analysis showed that, during vaccination, the RAP-treated rabbits were more confident with the surgeon and presented better adaptation than the non-treated ones. Over progressive clinical visits, which were conducted seven times, the RAP-treated rabbits were also less agitated than their non-treated counterparts. RAP seems to be a promising tool for helping rabbits during these veterinary procedures.

## 1. Introduction

The rabbit is the third most popular pet in families in several countries, and its diffusion is constantly increasing [1]. Domestic rabbits are usually exposed to several stressors and situations that necessitate adequate adaptation. Challenges may include exposure to a novel environment [2], inappropriate housing conditions [3,4], a lack of companionship [5,6], inappropriate feeding, and poor handling techniques and veterinary care [7,8,9,10]. Moreover, the rabbit is a prey by nature [11], and it continues to perceive humans as predators and their presence as a source of negative stimuli [12]. Stress is an adaptive biological phenomenon that occurs in animals to respond to changes in their environments [13] and is induced by a stimulus that triggers the activation of the hypothalamic–pituitary–adrenocortical (HPA) axis and the sympathetic–adrenomedullary system [14]. As a consequence of this, different physiological and behavioural changes occur, which guarantee the survival of the animals [15]. In the short term, the response to a stressor is triggered by the activation of the sympathetic nervous system, which releases catecholamines (norepinephrine and epinephrine) from the adrenal medulla [15]. During this phase, motor activity and contractions of the heart and spleen increase, releasing more red blood cells; vasomotor adjustments and pupil dilation occur, blood coagulation increases, and lymphocytes are neutralised to repair tissue damage, while energy is increased through glycolysis and lipolysis [16]. In the longer term, the corticotropin-releasing hormone, adrenocorticotropic hormone, and corticosteroids act thanks to the activation of the HPA [13]. Rabbits have some behavioural peculiarities due to the fact that they were domesticated later than other species, thus showing behaviours typical of wild rabbits, such as postpartum mating, maternal behaviour, nest building, neonatal reactions, and the social system [17,18]. This aspect makes this species a shy and independent animal [12], contrary to other domesticated species that have been selected for their docility and because they were not afraid of people [19]. Nevertheless, despite this late domestication, rabbits have significantly changed, with a decrease in their sensitivity to stress compared to their wild counterparts, supported by a reduction in the volume of the amygdala and an enlargement of the prefrontal cortex [20]. This evolution can facilitate their adaptation to interactions with humans, especially in the context of being a pet.

It is well known that veterinary visits can be stressful for pets; veterinary handling techniques need to be improved to improve pets’ welfare and, potentially, the well-being of veterinary staff, reducing the risk of injuries [21,22]. A clinical examination involves a series of situations and medical procedures that induce fear, stress, and anxiety in pets, such as standing in the waiting room, physical restraint and handling, and the use of needles for blood sampling or injections [21,22,23]. Among such procedures, vaccinations are an important source of behavioural and physiological stress in animals as well as in humans because they can cause pain in the injection zone, and they also induce inflammatory reactions needed to induce antibody production [24,25]. In companion animals, behavioural and physical changes such as lethargy, anorexia, and pyrexia are commonly observed after vaccination [24,26].

The use of pheromones to help animals cope with veterinary consultation stress has previously been evaluated in pets, and it has been proposed as a useful aid before and during a visit [21,22]. In dogs, the effect of dog appeasing pheromone (DAP)—a synthetic analogue of the maternal appeasing pheromone produced and secreted in the intermammary region by the nursing bitch and that has a reassuring effect on puppies and adults [27]—during veterinary procedures has also been explored [28,29]. More specifically, previous studies showed that DAP helped to reduce anxiety in dogs in the waiting and examination rooms [30,31] and improved the recovery and welfare of dogs subjected to surgical acts [32]. The spraying of feline facial pheromones has been shown to improve cats’ welfare during veterinary visits by reducing their stress [33] and to have calming effects in cats undergoing intravenous catheterisation with or without acepromazine [34]. Further research is needed to confirm the use of pheromones in handling these kinds of problems in pets. They are widely used in veterinary medicine and are proposed as helpful tools for improving animal welfare during veterinary procedures [22]. Moreover, they have other advantages, such as their safety, the absence of adverse effects and undesired drug interactions, and their ease of administration [31].

The synthetic analogue of rabbit appeasing pheromone (RAP) is based on a mixture of fatty acids, reproducing the composition of the original substance produced by the doe upon giving birth. Under natural circumstances, this substance ensures the attraction of the pups to the nest while being a reassuring message for them [35]. A preliminary study conducted on farm rabbits showed that RAP-treated animals appeared quieter, while improvements were observed for zootechnical parameters such as fertility, litter size, and viability at birth [36]. Another preliminary study showed that the use of RAP helped rabbits to better cope with novel environmental conditions [37]. Appeasing pheromones have also been isolated in other species, such as pigs, horses, cattle, cats, and chickens, and their use has been associated with reductions in stress marker levels and anxiety as well as improvements in animal welfare [38,39,40,41].

The aim of this study was to evaluate, for the first time, whether continuous exposure to RAP could help rabbits cope with the stress induced by routine clinical consultations and vaccinations with the anti-myxomatosis and rabbit haemorrhagic disease virus vaccine, which is commonly used in this species. Moreover, during this study, a new diffuser technology was used for the first time to administer the appeasing pheromone: a device composed of stick diffusers that allow a constant release of RAP into the rabbits’ living environment (patent pending).

## 2. Materials and Methods

This study was performed at the experimental facility of the Research Institute in Semiochemistry and Applied Ethology (IRSEA, France, 43°52′37″ nord, 5°23′49″ est). This study was conceived and performed in strict accordance with French (2013-118) and European laws (2010/63/EU) on the protection of animals used for scientific purposes. The study protocol was approved by the Ethics Committee “Laurent Vinay” (CEEA-071) and the French Ministry of Research (APAFIS #27008-2020080716514330 v6).

### 2.1. Animals and Housing

Twenty-four 5-week-old New Zealand white rabbits (Charles River labs, Lyon, France) were included in the study and randomly divided by sex and weight into two groups of 12 individuals (6 males and 6 females). The groups were housed in two separate, identical and independent experimental rooms under stable environmental conditions: a mean ambient room temperature of 21.5 °C, a mean relative humidity of 53%, and a natural photoperiod (12–12 h daylight). Two rabbits of the same sex were housed per cage (placed upon the soil, sized L96 cm × P57 cm × H56 cm, Casita 100, Ferplast, Illkirch-Graffenstaden, France). Each cage contained two platforms (which enabled the rabbits to hide), two feeders, two 1-litre nipple drinkers, absorbent litter (Copo-Box^®^, SICSA, Courbehaye, France) and, as enrichment materials, a wood stick and four chewable toys. A standard rabbit pellet diet (SERLAB, Montataire, France) was provided at 400 g/cage/day, and water was provided *ad libitum*. No antibiotics were administered to animals via water or food. During their first 5 weeks spent in the Charles River Labs facilities, the rabbits were submitted to their classical “Rabbit Breeding Programme”, which also includes a handling program designed in two phases, from their birth until their departure. When arrived at our facility, we followed Charles River instructions, which recommend petting the rabbit in the operator’s arms for a minimum of two minutes, twice a day, throughout the acclimatisation period. If the rabbit is too stressed, petting can be offered directly in the cage.

### 2.2. Treatment

The rabbits were randomly assigned to RAP treatment and placebo rooms by the Statistical Service. The field operators worked in a blinded fashion throughout the study and data collection periods. The treatment (SignsLab, Apt, France) was provided using a glass container filled with 40 mL of a solution of 2% RAP diluted in a mixture of complex alcohol and antioxidants and six natural rattan sticks that were 24 cm long (Figure 1).

The rabbits in the control group were housed in the other room and administered only the excipient using the same device. Both the RAP treatment and control containers were placed in the housing rooms two days before the rabbits’ arrival to allow diffusion into the air. Two devices with six sticks each were placed in each room, one per side, at the bottom and at the entrance, at the same level as the cages. According to the manufacturer’s instructions, the devices providing the treatment and the control were replaced every ten days.

### 2.3. Experimental Design

The study on animals began on 13 December 2023 and ended on 25 January 2024. This study was conducted over 43 days, starting from the day on which the 5-week-old rabbits arrived, at which point they were weighed to allow randomisation (D0). Five days later (D5), the animals were weighed and subjected to a routine clinical consultation and blood sampling (2 mL) to confirm that they were all in good health. The blood was sampled from the saphenous vein, during which the rabbit was gently handled, keeping its head under the operator’s hand to expose the posterior leg toward the surgeon who realised the sampling. After the consultation, the same operator used a visual analogue scale (VAS) to assess the rabbits’ reactivity and state (Appendix A) during the visit handling.

Eight days after arrival (D8), the rabbits were vaccinated with Nobivac Myxo-RHD PLUS (MSD Animal Health, Boxmeer, The Netherlands), which offers double protection against the myxoma virus and the rabbit haemorrhagic disease virus (RHDV), according to the manufacturer’s instructions. Before the vaccination, the rabbits were subjected to a clinical consultation to fill out the VAS. The VAS and videos were also used to assess each rabbit’s reactivity and state during the vaccination. On days 15, 22, 29, 36, and 43 (D15, D22, D29, D36, and D43), the routine clinical consultation was repeated to fill out the VAS. The VAS was always filled out by the same operator, a veterinarian surgeon and resident of the European College of Animal Welfare Behavioural Medicine and a member of the Animal Welfare Body (E.C.), to ensure that the results were comparable across days. Figure 2 shows the plan for the development of the study.

The clinical consultation consisted of body weighing; examination of the external mucosae, skin, ears, eyes, and mouth; paw and abdomen palpation; and heart and lung auscultation. On D5, blood sampling was also performed.

### 2.4. Analysed Parameters

The VASs (Appendix A) were used to measure the rabbits’ reactivity and state during the clinical examination. On D8, the rabbits’ reactivity and state were also measured during the vaccination by using the VAS (Appendix A). The act of vaccination was video-recorded (GoPro Hero 10, San Mateo, CA, USA) to enable the behaviours listed in the ethogram in Table 1 to be assessed, and the videos were analysed by two independent operators (E.C. and J.D.). The weights and average daily gains (ADGs), measured on days 0, 5, 8,15, 22, 29, 36, and 43, were also included in the statistical analysis.

### 2.5. Statistical Analysis

The data were analysed using R version 4.1.2 (2021-11-01) software and RStudio version 2021.09.1 + 372 © (R Foundation for Statistical Computing, Vienna, Austria). The significance threshold was fixed at the classical value of 5%.

For the VAS analysis during the vaccinations, continuous variables related to the reactivity and states of the rabbits during vaccination were analysed using General Linear Mixed Models (GLMMs) and the lme4 package in R. The effect of treatment (A and B) was included in the models as a fixed effect. The conditions of residual normality (graphically and using normality tests) and homoscedasticity (using a “Residuals versus fits” graphic and Levene’s test) were validated. For the VAS analysis during the clinical examination, continuous variables related to the reactivity and states of the rabbits during the clinical examinations were analysed using General Linear Mixed Models (GLMMs) and the lme4 package in R. The effects of treatment (A and B), day (5, 8, 15, 22, 29, 36 and 43), and treatment × day interaction were included in the models as fixed effects. The condition of residual normality (graphically and using normality tests) was validated for the two parameters (after transformation for the reactivity). Homoscedasticity (using a “Residuals versus fits” graphic and Levene’s test) was validated for the reactivity of the rabbits. For the states, a model considering the heterogeneity of variances was used.

For video analysis, first, the interobserver reliability between the two independent readers was assessed for each parameter measured from the video. The normality of each variable and each video reader was verified using normality tests (Shapiro–Wilk) and QQ-plot graphs. If the variable followed a normal distribution for the two video readers, Pearson’s correlation coefficient was used. If normality was not checked for at least one video reader, Spearman’s correlation coefficient was preferred. The association between the two readers was computed by squaring the correlation coefficient obtained. If the association was greater than 80%, the mean of the two video readers was computed and used for the rest of the analysis. If the association was lower than 80%, the variable was reworked by the readers.

After the interobserver reliability had been assessed, discrete variables related to the occurrence of behaviours were analysed using Generalized Linear Mixed Models (GzLMMs) for counting data using the Poisson distribution and the lme4 package. The effect of treatment (A and B) was included in the models as a fixed effect. The Pearson chi-square/DF statistic was used to evaluate the dispersion of the data. No overdispersion was detected for any of the parameters.

The durations of sitting or standing, movement, ear erected, and ear middle/flattened were analysed using General Linear Mixed Models (GLMMs) using the lme4 package in R. The effect of treatment (A and B) was included in the models as a fixed effect. The conditions of residual normality (graphically and using normality tests) and homoscedasticity (using the “Residuals versus fits” graphic and Levene’s test) were validated, except for the duration of movement, for which a Box–Cox transformation was applied to the data, enabling the model to be run on these transformed data.

The durations of flattened ears and contact with the veterinarian were analysed using hurdle mixed models because many zeros were found, which implied that a classical model did not fit. The package used for this model was GLMMadaptive. The effect of treatment (A and B) was included in the models as a fixed effect. The condition of residual normality (graphically and using normality tests) for the positive part of the model was validated for these two parameters.

For all the studied parameters, multiple comparisons were performed using the Tukey test for effects with more than two modalities. Complete models were simplified if the AIC and BIC criteria decreased when an effect was removed from the models (and if this effect was not significant).

Finally, in order to analyse the weights of the rabbits and average daily gain (continuous variables), General Linear Mixed Models (GLMMs) were realised using the lme4 package in R. The effects of treatment (A and B) and day (5, 8, 15, 22, 29, 36 and 43) were included in the models as fixed effects. A baseline was added to the model as a covariate for the weight (weight of rabbits on arrival). The conditions of residual normality (graphically and using normality tests) and homoscedasticity (using the “Residuals versus fits” graphic and Levene’s test) were verified and validated.

## 3. Results

### 3.1. Vaccination

The interobserver reliability between the two observers who performed the video analysis was calculated as in Table 2. The statistical analysis showed that, during the vaccination, the RAP-treated rabbits were more adapted to the situation than the controls (GLMM; *p* = 0.02) according to the rabbit state VAS observations (Figure 3). No difference in rabbit reactivity was observed (GLMM; *p* = 0.14). The video analysis also revealed that the RAP group spent significantly more time moving on the table (GLMM; *p* = 0.03) and more time in contact with the veterinarian after the vaccination (hurdle model; *p* = 0.039; Figure 4). Moreover, the control rabbits kept their ears in the middle/flattened position more frequently (Poisson model; *p* = 0.04) and for a longer time (GLMM; *p* = 0.003) than the RAP-treated ones.

Concerning the rabbit positions (flattened and sitting/standing), no differences in either frequency or duration were observed.

### 3.2. Clinical Consultation

The VAS analysis showed no differences between the RAP and control groups concerning the treatment’s effect on the rabbits’ state (GLMM; *p* = 0.4) or reactivity (GLMM; *p* = 0.57) during the clinical examination. A treatment × day effect was observed for both parameters. The control rabbits were more agitated during the consultation, and the scores worsened during the study (GLMM; *p* < 0.0001; Figure 5). Concerning the rabbits’ reactivity (GLMM; *p* < 0.0001), the RAP group became less reactive during the consultations throughout the study. Finally, no differences between the two groups were observed concerning weight (GLMM; *p* = 0.42) and ADG (GLMM; *p* = 0.42). Moreover, for all tested parameters, no differences were observed between male and female rabbits. The results of the clinical consultation VAS and weight are reported in Table 3.

## 4. Discussion

This study evaluated, for the first time, the effects of the rabbit-appeasing pheromone in helping rabbits cope with the fear and stress induced by routine veterinary procedures such as vaccinations and clinical consultations. Even if further studies are needed before firmer conclusions can be drawn, our findings seem to suggest that continuous exposure to RAP, realised using a passive diffusion system that was employed for the first time, can help pet rabbits better adapt to some stressful situations linked to veterinary care, as previously shown in cats and dogs [30,31,32,33,34]. Compared to the animals in these previous studies, the rabbits in the present study were continuously exposed to the appeasing pheromone in their living environment, not just during or shortly before veterinary procedures.

In particular, during vaccination or in the moments right after this stressful act, RAP treatment seemed to make the rabbits more confident; they were less fearful and inhibited, as shown by their state and tendency to seek contact with the veterinarian and move to explore the table surface. By contrast, non-treated rabbits moved less, did not seek human contact and kept their ears pushed back, which is commonly associated with a negative emotional state and is a known sign of submission, pain, distress and, more generally, poor welfare in rabbits [43,44,45,46]. Vaccination is known to be stressful in all species [24,25] because it can induce pain in the injection zone and also requires robust physical restraint to avoid animal movement during the injection. The effects observed in our study may suggest that a rabbit constantly exposed to the appeasing pheromone in its living environment is able to better adapt to and cope with a stressful situation, probably because it is less inhibited and fearful.

Additionally, during the sequence of clinical examinations, RAP treatment seemed to help rabbits cope with this situation. The statistical analysis of the VAS scores assessing both the rabbits’ state and reactivity showed a significant difference in the treatment × day effect. These findings suggest that, from the days after the vaccination and throughout the course of the study, the control rabbits became less confident and more agitated during the routine veterinary procedures, while the RAP-treated animals appeared slightly, but significantly, less stressed. In particular, the rabbit state scores strongly decreased in the control group, and they were also less stable, with important variations among the different days. Except for day 29, when the RAP group had a temporary decrease in the scores, probably due to the malfunctioning of our facility that induced an environmental change in temperature and humidity, the RAP group had more constant scores, probably suggesting a modulatory role of the RAP in rabbits’ emotional state, allowing the treated rabbits to be more confident despite the repetition of these stressful procedures. In fact, as stated by previous authors, the clinical examination comprises a series of different acts that can alter an animal’s mental state and require it to continuously adapt, such as a temporary change in the environment, a succession of different handling techniques, and physical restraint [21,22,23]. Even if these findings need to be confirmed with further research, they suggest that constant exposure to RAP can help rabbits cope with the sequences of these routine clinical procedures, probably because RAP-treated animals tend to be less scared and more confident in facing these procedures. In rabbits, the use of medications to reduce stress in the veterinary clinic has been previously proposed with interesting results, in particular the administration of gabapentin, an analogue of the neurotransmitter gamma-aminobutyric acid [47]. Even if this drug is generally well tolerated, some side effects have been reported, including lethargy, sedation, and ataxia [48,49]. On the other hand, the use of pheromones has the advantages of being safe, having no adverse effects and undesired drug interactions, and being easy to administer [31]. Moreover, its effect is not based on a sedative action; thus, the rabbit can remain vigilant and interact with and explore the surrounding environment.

In our study, no differences were observed between male and female rabbits for all the tested parameters, probably due to the young age of the rabbits: they arrived when they were 5 weeks old, and the study was completed when they were 11 weeks old. In fact, it is known that the behavioural repertoire of rabbits towards humans may be influenced by the sex of the subjects during adulthood because of sexual hormones [50]. The rabbits in our study were far from adulthood and sexual maturity, both of which are considered to begin when the subjects are 1 year old [51]. This condition may easily explain why no difference was observed between males and females during our study.

Even though this study only focused on the previously mentioned veterinary procedures, it is important to consider that the 5-week-old rabbits included in this study were also exposed to two other stressors: transport from the rearing facility to our facilities and a novel environment [2]. Taking this into account, the RAP devices were placed in the experimental rooms two days before the rabbits’ arrival so that the pheromone would have already diffused into the air when the rabbits were placed in their novel environment in order to evaluate the efficacy of its continuous administration. It is the authors’ opinion that the immediate exposure to the RAP treatment upon arrival might have allowed for earlier protection from these stressors, thereby contributing to the modulation of the rabbits’ states throughout the course of the study, as shown by the effects during the vaccinations and clinical examinations. Nevertheless, further studies focused on these other stressful situations (transport and arrival in a novel environment) should be carried out to assess RAP’s efficacy in their handling.

The role of appeasing pheromones in facilitating the adaptation process has previously been assessed in many species and conditions, including in pets, farm animals, and wildlife animals. In dogs, many studies have shown that DAP can effectively treat travel-related problems, separation-related problems, and sound sensitivity, facilitate adaptation to a novel environment, and improve dog welfare in the shelter [29,52] beyond the effects of the clinical procedures previously mentioned. Cat appeasing pheromone (CAP) can be used to handle problems linked to the agonistic behaviour in domestic cats and zoo European wildcats [53,54,55]. In farm animals, the use of appeasing pheromones has been associated with reductions in stress marker levels and aggression and improvements in zootechnical performance and welfare [36,39,40,41,56,57]. Taken together, these findings highlight the ability of appeasing pheromones to help animals cope with challenging situations of different natures by acting on various behavioural and physiological levels and permitting better adaptation to these stressful conditions. Interestingly, these studies tested methods of administration that differed from a technological point of view (i.e., diffuser bloc, topical skin application, sprays, and collars) and in the duration of application (individual or continuous), depending on the tested situation. In the present study, we tested, for the first time, a new technology for diffusing an appeasing pheromone into the environment. RAP was diffused using a system that is commonly used at home (rattan sticks) and that allows a continuous and homogeneous release of the semiochemicals, specifically adapted for pets. Moreover, it is a passive diffusion system that does not require electricity or heat for the substance to be released. Previous studies involving continuous diffusion technologies (blocs, topical application, or electrical diffusers) showed that the constant diffusion of a maternal pheromone could benefit animals by modulating and improving their mental states, therefore permitting easier adaptation when facing challenging events.

## 5. Conclusions

Our study evaluated, for the first time, the effects of the continuous administration of rabbit appeasing pheromone in rabbits subjected to common veterinary procedures such as vaccinations and clinical examinations. Even if further research is needed to draw firmer conclusions, our findings seem to suggest that RAP can help rabbits cope with these veterinary acts, permitting them to better adapt, reducing their stress, and thus improving their overall welfare, as observed in other companion animals. Since the rabbit is the third most popular pet in families in various countries [1], this treatment seems to be a promising tool for improving the welfare of pet rabbits at different stages of their lives.

## Figures and Tables

**Figure 1 animals-14-03549-f001:**
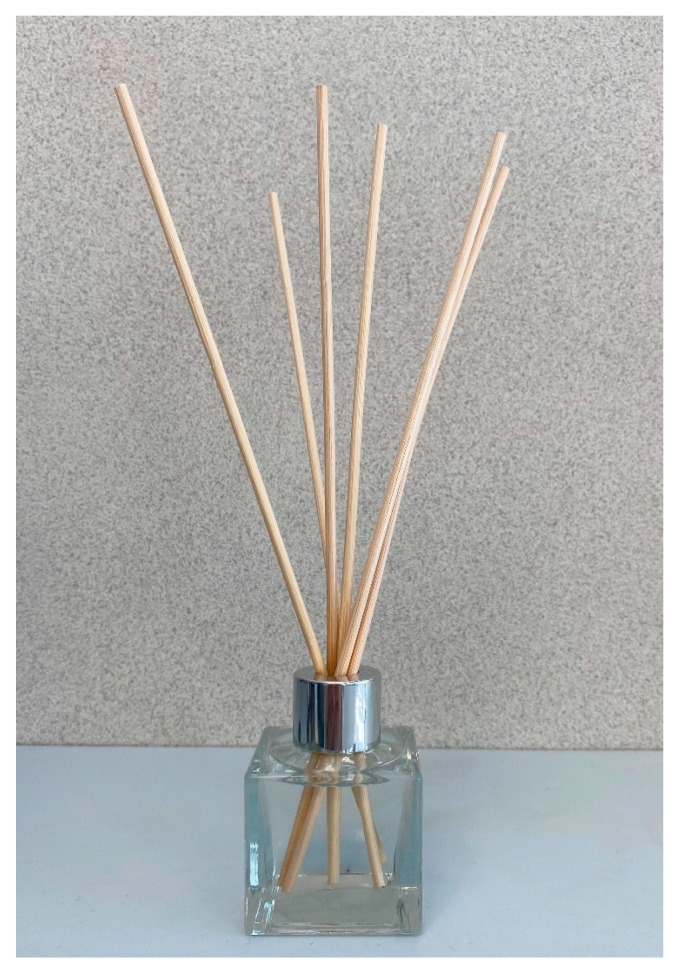
The device used to diffuse the RAP and the excipient in the experimental rooms.

**Figure 2 animals-14-03549-f002:**
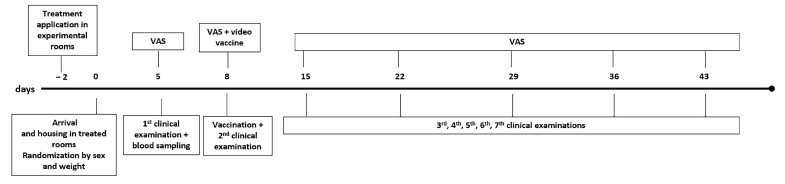
Development of the study from treatment application until the end.

**Figure 3 animals-14-03549-f003:**
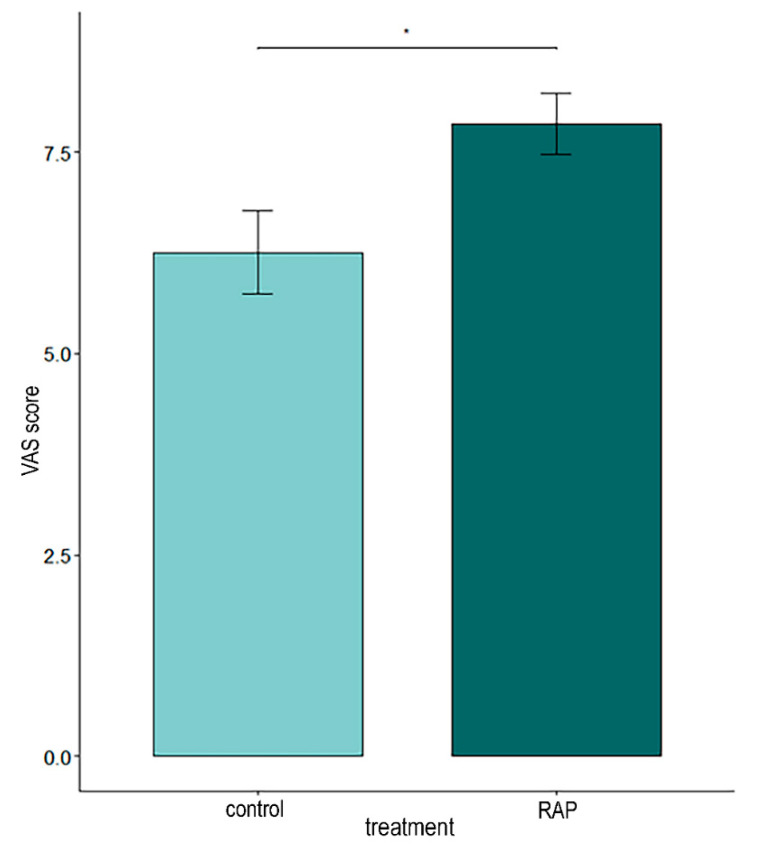
Rabbit state scores in the control and RAP groups during vaccination (* *p* < 0.05).

**Figure 4 animals-14-03549-f004:**
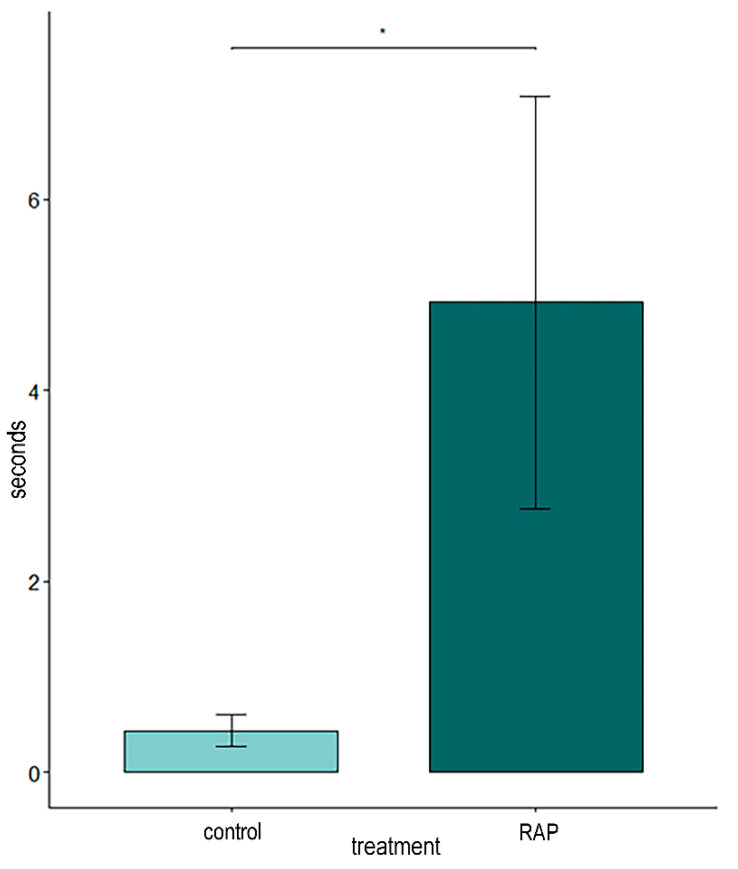
Comparison of duration of contact with the veterinarian (in seconds) between the control and RAP groups during vaccination (* *p* < 0.05).

**Figure 5 animals-14-03549-f005:**
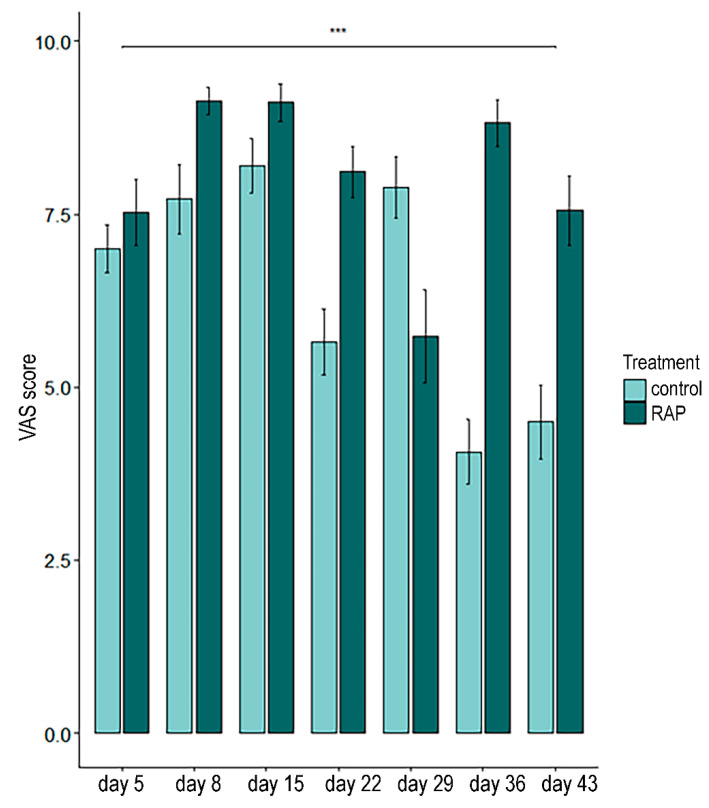
Rabbit state scores during the visits throughout the study in the control and RAP groups (*** *p* < 0.01).

**Table 1 animals-14-03549-t001:** Ethogram of rabbits during vaccination; adapted from previous studies [42,43,44].

Behaviour	Description	Type of Measurement
Flattened	The rabbit is in a flattened position; the abdomen touches the table, and four or two legs are spread.	Frequency Duration in seconds
Sitting or standing	Sitting: the rabbit is in a sitting position, with its hindlegs unstretched and on the ground, while its forelimbs are stretched with the paws touching the ground. Standing: the rabbit stands with its fore- and/or hindlegs unstretched and on the ground.	Frequency Duration in seconds
Movement	The rabbit moves with its fore- and/or hind legs or does a circle. The rabbit does an incomplete movement (change in position) or just attempts to move. The movements and/or movement attempts can have an explorative purpose, and this is shown by sniffing behaviour (the air or the table) and/or moving the head around (bobbing head).	Frequency Duration in seconds
Ear erected	The ears are erected, perpendicular to the spine.	Frequency Duration in seconds
Ear middle/flattened	The ears are positioned behind from 45° until completely adherent to the back.	Frequency Duration in seconds
Post-vaccination contact with the veterinarian	The rabbit touches the veterinarian (clothes, hands, etc.).	Duration in seconds

**Table 2 animals-14-03549-t002:** Interobserver reliability between the two observers who performed the video analysis.

Behaviour	Type of Measurement	Correlation Coefficient Used	Correlation Coefficient	Association
Flattened	Frequency Duration in seconds	Spearman	0.9967	99.3%
		Spearman	0.9970	99.4%
Sitting or standing	Frequency Duration in seconds	Spearman	0.9272	86.0%
		Pearson	0.9880	97.6%
Movement	Frequency Duration in seconds	Spearman	0.9682	93.7%
		Spearman	0.9870	97.4%
Ear erected	Frequency Duration in seconds	Spearman	0.9039	81.7%
		Pearson	0.9105	82.9%
Ear middle/flattened	Frequency Duration in seconds	Spearman	1.0000	100.0%
		Spearman	1.0000	100.0%
Post-vaccination contact with the veterinarian	Duration in seconds	Spearman	0.9954	99.1%

**Table 3 animals-14-03549-t003:** Results of the statistical analysis (GLMM) of rabbit behaviour during the clinical examination.

Parameter	Effect	Mean + SD	*p*-Value
Rabbit state by VAS	Treatment	Control: 6.45 ± 2.17 RAP: 7.99 ± 1.83	0.4014
	Treatment × day	Control D5: 7.00 ± 1.22 D8: 7.72 ± 1.73 D15: 8.20 ± 1.38 D22: 5.65 ± 1.63 D29: 7.88 ± 1.54 D36: 4.07 ± 1.61 D43: 4.50 ± 1.78 RAP D5: 7.53 ± 1.67 D8: 9.13 ± 0.67 D15: 9.12 ± 0.94 D22: 8.11 ± 1.29 D29: 5.73 ± 2.31 D36: 8.82 ± 1.12 D43: 7.55 ± 1.75	<0.0001
Rabbit reactivity by VAS	Treatment	Control: 8.26 ± 2.05 RAP: 8.06 ± 2.00	0.5684
	Treatment × day	Control D5: 9.17 ± 0.74 D8: 9.09 ± 1.00 D15: 9.10 ± 0.90 D22: 6.28 ± 2.10 D29: 8.59 ± 0.77 D36: 6.64 ± 3.28 D43: 9.05 ± 1.83 RAP D5: 9.09 ± 1.34 D8: 7.16 ± 0.89 D15: 7.12 ± 1.56 D22: 7.16 ± 2.51 D29: 7.83 ± 2.43 D36: 9.13 ± 1.59 D43: 9.04 ± 2.06	<0.0001
Body weight (kg)	Treatment	Control: 1.86 ± 0.48 RAP: 1.83 ± 0.45	0.4206
	Treatment × day	Control D5: 1.23 ± 0.12 D8: 1.34 ± 0.14 D15: 1.63 ± 0.15 D22: 1.89 ± 0.17 D29: 2.09 ± 0.18 D36: 2.31 ± 0.22 D43: 2.52 ± 0.24 RAP D5: 1.24 ± 0.11 D8: 1.34 ± 0.12 D15: 1.61 ± 0.13 D22: 1.83 ± 0.13 D29: 2.05 ± 0.15 D36: 2.28 ± 0.17 D43: 2.47 ± 0.17	<0.0001
Average daily gain (kg/day)	Treatment	Control: 0.04 ± 0.1 RAP: 0.03 ± 0.1	0.4228

## Data Availability

The data are available upon request.

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
