# Peer review of "Helping Rabbits Cope with Veterinary Acts and Vaccine-Related Stress: The Effects of the Rabbit Appeasing Pheromone (RAP)"

_animals, 2024, doi:10.3390/ani14233549_

Round 1

Reviewer 1 Report

Comments and Suggestions for Authors

Really innovative work properly described with good results that can be useful in daily working practice with pet rabbits. Looking foreward for new upgrades about the RAP use in other stressful situation. 

RAP can help pet rabbits to cope with a stressful situation as veterinary visit and vaccination as well as post vaccination related stress.

I think that this study is really interesting and relevant for veterinarians involved in exotic pet medicine and in particular in rabbit medicine because rabbits are prey animals and they are well known to be stressed for external factors. In this study the way the RAP rabbits interact with the environment and with the veterinarian is really surprising as well as the absence of fear reaction afterwards.

As far as I know there are not published material about it in pet rabbits.

Authors may consider in the next trials stressful situation as the travel to the clinic as well as hospitalization.
Moreover another stressful factor that can be considered is the noises and the odors from cats and dogs in a veterinary clinic.

Conclusion exposed are clear and address the main question explaining that RAP rabbits were more calm and showed less uncomfortable reactions. Moreover authors considered and underlined that further research are needed to add more knowledge to the topic and better understand the advantages of RAP.
An important key-point that is well explained is that in this study rabbits were continuously exposed to RAP instead of short term treatment before the induced stressful situation.

Author Response

Really innovative work properly described with good results that can be useful in daily working practice with pet rabbits. Looking foreward for new upgrades about the RAP use in other stressful situation. 

RAP can help pet rabbits to cope with a stressful situation as veterinary visit and vaccination as well as post vaccination related stress.

I think that this study is really interesting and relevant for veterinarians involved in exotic pet medicine and in particular in rabbit medicine because rabbits are prey animals and they are well known to be stressed for external factors. In this study the way the RAP rabbits interact with the environment and with the veterinarian is really surprising as well as the absence of fear reaction afterwards.

As far as I know there are not published material about it in pet rabbits.

Authors may consider in the next trials stressful situation as the travel to the clinic as well as hospitalization.
Moreover another stressful factor that can be considered is the noises and the odors from cats and dogs in a veterinary clinic.

Conclusion exposed are clear and address the main question explaining that RAP rabbits were more calm and showed less uncomfortable reactions. Moreover authors considered and underlined that further research are needed to add more knowledge to the topic and better understand the advantages of RAP.
An important key-point that is well explained is that in this study rabbits were continuously exposed to RAP instead of short term treatment before the induced stressful situation.

Thank you very much for your kind comments on our work. Concerning the trials that should be done about other important stressful situations, thank you for your suggestions. We have already planned some of the future studies on those directions.

Reviewer 2 Report

Comments and Suggestions for Authors

It should be said that the authors have undertaken very interesting research in the field of animal welfare, particularly stress in animals during veterinary procedures and many grooming procedures. Thus, the study deals with very interesting research in improving the welfare of captive animals, which perfectly aligns with the current trend of research in animal behaviour and welfare. Although the study did not cover all the stressful situations that can accompany animals (rabbits) with veterinary treatments and procedures, this research provides a start for further work in this area, which the authors suggest.

Two minor comments occurred to me that need clarification:

For keywords, I would remove the single phrase rabbit, as it is used in another keyword, so there is no point in duplicating it.

Figure 5, it is worth explaining or at least hypothesizing, what was the reason for the much greater variation in the evaluation of the condition of the rabbits in the control group, as well as the situation on day 29 of the study?

I leave it to the Academic Editor to decide whether to take my comments into account, and I recommend the manuscript for publication.

Author Response

It should be said that the authors have undertaken very interesting research in the field of animal welfare, particularly stress in animals during veterinary procedures and many grooming procedures. Thus, the study deals with very interesting research in improving the welfare of captive animals, which perfectly aligns with the current trend of research in animal behaviour and welfare. Although the study did not cover all the stressful situations that can accompany animals (rabbits) with veterinary treatments and procedures, this research provides a start for further work in this area, which the authors suggest.

Thank you very much for your kind words.

Two minor comments occurred to me that need clarification:

For keywords, I would remove the single phrase rabbit, as it is used in another keyword, so there is no point in duplicating it.

Thank you for your suggestion, we followed it and deleted the single word “rabbits”.

Figure 5, it is worth explaining or at least hypothesizing, what was the reason for the much greater variation in the evaluation of the condition of the rabbits in the control group, as well as the situation on day 29 of the study?

 Thank you for this comment. We partially modified the discussion of these data (lines 322-328) and we also add a possible explanation of results on Day 29. In fact, checking on our records, that week our facility faced a problem in the ventilation system  that mainly affects the T° and the Humidity of the room of the RAP group.

I leave it to the Academic Editor to decide whether to take my comments into account, and I recommend the manuscript for publication.

Reviewer 3 Report

Comments and Suggestions for Authors

Introduction

To reflect the importance of the rabbit as a companion animal.

A definition of stress should be included, indicating that it is an adaptive phenomenon in the response of an animal to changes in its environment (Veissier and Boissy, 2007) and involves the organism's response to a stimulus that triggers the activation of the hypothalamic-pituitary-adrenocortical (HPA) axis and the sympathetic-adrenomedullary system (Möstl and Palme, 2002). For example, it could be stated as follows:

Stress is an adaptive phenomenon in the response of an animal to changes in its environment (Veissier and Boissy, 2007) and involves the organism's response to a stimulus that triggers the activation of the hypothalamic-pituitary-adrenocortical (HPA) axis and the sympathetic-adrenomedullary system (Möstl and Palme, 2002). Different studies have shown that this occurs by triggering a series of behavioural changes, mainly physiological and escape (Temple et al., 2014). These physiological alterations are rapid responses that guarantee the survival of the animals (Wingfield et al., 1997). In the short term, a response to a stressful stimulus is triggered by the activation of the sympathetic nervous system, which releases catecholamines (norepinephrine and epinephrine) from the adrenal medulla (Cunningham, 1999). According to Axelrod and Reisine (1984), motor activity and contractions of the heart and spleen increase in this phase, releasing more red blood cells. Vasomotor adjustments and pupil dilation occur, blood coagulation increases and lymphocytes are neutralised to repair tissue damage, while energy is increased through glycolysis and lipolysis (Axelrod and Reisine, 1984). In the longer term, the corticotropin-releasing hormone, adrenocorticotropic hormone and corticosteroids (mainly corticosterone and cortisol in rabbits) act through the HPA (Kataoka et al., 2014).

It should be noted that rabbits continue to perceive humans as predators and are predisposed to associate their presence with negative stimuli, which is often a factor of stress and fear (Trocino and Xiccato, 2006), since they are prey animals by nature (Benato et al., 2019). It should be added that shyness is one of the main attributes of rabbits, being elusive and independent animals (Trocino and Xiccato, 2006), which makes it more difficult to perceive their fear or acute stress. All this is contrary to the preferential characteristics that are selected when domesticating animals, such as docility and not being afraid of people (Parsons, 1988). These behavioural characteristics of rabbits may be due to the fact that they were domesticated much later than other species (Morton, 2002) and the effects of domestication on them are not as marked, with many characteristics of their wild ancestors still being preserved, such as digging burrows and making nests (Stodart and Myers, 1964; EFSA, 2005). Also, it should be noted that, although they can be calm and docile animals, they tend to be skittish (Ward, 2006).

Reflecting that stress can also cause complications such as gastrointestinal hypomotility

Reflecting that there are medications to reduce stress in the veterinary clinic, e.g.:

Conway RE, Burton M, Mama K, Rao S, Kendall LV, Desmarchelier M, Sadar MJ. 2023. Behavioral and Physiologic Effects of a Single Dose of Oral Gabapentin in Rabbits (Oryctolagus cuniculus). Top Companion Anim Med., Mar-Jun; pp 53-54: 100779.

2. Materials and methods

The authorisation to carry out animal experiments must be sent to the editor.

2.1 Animals and housing

Include geographic coordinates of the test site.

Also indicate the presence or absence of antibiotics in the feed or water.

Amount of water supplied or given ad libitum.

Indicate whether or not the animals have been previously subjected to handling that can reduce stress according to the following bibliography:

Bilkó Á, Altbäcker V, 2000. Regular handing early in the nursing period eliminates

fear responses toward human beings in wild and domestic rabbits. Dev Psychobiol

36 (1): 78-87. https://doi.org/10.1002/(SICI)1098-2302(200001)36:1<78::AIDDEV8>

3.0.CO;2-5

Csatádi K, Kustos K, Eiben C, Bilkó Á, Altbäcker V, 2005. Even minimal human

contact linked to nursing reduces fear responses toward humans in rabbits. Appl

Anim Behav Sci 95 (1-2): 123-128. https://doi.org/10.1016/j.applanim.2005.05.002

2.3. Experimental design

Indicate the start and end date of the trial.

State the weights of the males and females, as well as the differences detected throughout the study.

Where was the blood sample obtained from and with what material? How were the rabbits collected? Please indicate the handling method used to prevent injuries.

What parameters were analysed in the blood? In which laboratory? Is it accredited according to ISO 17025 or does it have a certified quality system?

Indicate the reference bibliography for the analytical parameters in blood.

What video camera was used? Indicate brand and model

Were the rabbits previously dewormed?

3. Results

It is important to differentiate the results between males and females or adequately justify why this was not done.

The average weights in each of the weighings must be reflected in a table, differentiating females and males

The levels of the parameters measured in blood on each of the dates must be reflected in a table, differentiating females and males.

4. Discussion

Indicate the advantages of using pheromones compared to medications to relieve or prevent stress.

Author Response

Introduction

To reflect the importance of the rabbit as a companion animal.

Thank you for your suggestion, we added this at the beginning of the introduction. (lines 54-55)

A definition of stress should be included, indicating that it is an adaptive phenomenon in the response of an animal to changes in its environment (Veissier and Boissy, 2007) and involves the organism's response to a stimulus that triggers the activation of the hypothalamic-pituitary-adrenocortical (HPA) axis and the sympathetic-adrenomedullary system (Möstl and Palme, 2002). For example, it could be stated as follows:

Stress is an adaptive phenomenon in the response of an animal to changes in its environment (Veissier and Boissy, 2007) and involves the organism's response to a stimulus that triggers the activation of the hypothalamic-pituitary-adrenocortical (HPA) axis and the sympathetic-adrenomedullary system (Möstl and Palme, 2002). Different studies have shown that this occurs by triggering a series of behavioural changes, mainly physiological and escape (Temple et al., 2014). These physiological alterations are rapid responses that guarantee the survival of the animals (Wingfield et al., 1997). In the short term, a response to a stressful stimulus is triggered by the activation of the sympathetic nervous system, which releases catecholamines (norepinephrine and epinephrine) from the adrenal medulla (Cunningham, 1999). According to Axelrod and Reisine (1984), motor activity and contractions of the heart and spleen increase in this phase, releasing more red blood cells. Vasomotor adjustments and pupil dilation occur, blood coagulation increases and lymphocytes are neutralised to repair tissue damage, while energy is increased through glycolysis and lipolysis (Axelrod and Reisine, 1984). In the longer term, the corticotropin-releasing hormone, adrenocorticotropic hormone and corticosteroids (mainly corticosterone and cortisol in rabbits) act through the HPA (Kataoka et al., 2014).

It should be noted that rabbits continue to perceive humans as predators and are predisposed to associate their presence with negative stimuli, which is often a factor of stress and fear (Trocino and Xiccato, 2006), since they are prey animals by nature (Benato et al., 2019). It should be added that shyness is one of the main attributes of rabbits, being elusive and independent animals (Trocino and Xiccato, 2006), which makes it more difficult to perceive their fear or acute stress. All this is contrary to the preferential characteristics that are selected when domesticating animals, such as docility and not being afraid of people (Parsons, 1988). These behavioural characteristics of rabbits may be due to the fact that they were domesticated much later than other species (Morton, 2002) and the effects of domestication on them are not as marked, with many characteristics of their wild ancestors still being preserved, such as digging burrows and making nests (Stodart and Myers, 1964; EFSA, 2005). Also, it should be noted that, although they can be calm and docile animals, they tend to be skittish (Ward, 2006).

Reflecting that stress can also cause complications such as gastrointestinal hypomotility

We thank the reviewer for his/her suggestion. Anyway, we decided to add to our manuscript only a brief but important part about the rabbit as a prey (59-61), since the study was not focused on the physiological ways of the stress, but only on a specific type of stress that occurs in pets in particular circumstances. Moreover, the study did not explore physiological parameters, but only behavioural responses. 

Reflecting that there are medications to reduce stress in the veterinary clinic, e.g.:

Conway RE, Burton M, Mama K, Rao S, Kendall LV, Desmarchelier M, Sadar MJ. 2023. Behavioral and Physiologic Effects of a Single Dose of Oral Gabapentin in Rabbits (Oryctolagus cuniculus). Top Companion Anim Med., Mar-Jun; pp 53-54: 100779.

Thank you for this suggestion, which is very interesting. We decided to added this part all in the discussion, as you can see in lines 335-343.

  1. Materials and methods

The authorisation to carry out animal experiments must be sent to the editor.

Thank you for the suggestion. We have asked to the editor how to send it to the editors.

2.1 Animals and housing

Include geographic coordinates of the test site.

Thank you for this suggestion, we added this information at the beginning of the paragraph 2 (line 109)

Also indicate the presence or absence of antibiotics in the feed or water.

Amount of water supplied or given ad libitum.

Thank you for these suggestions, we added these information at the end of the paragraph 2.1 (lines 125-126)

Indicate whether or not the animals have been previously subjected to handling that can reduce stress according to the following bibliography:

Bilkó Á, Altbäcker V, 2000. Regular handing early in the nursing period eliminates

fear responses toward human beings in wild and domestic rabbits. Dev Psychobiol

36 (1): 78-87. https://doi.org/10.1002/(SICI)1098-2302(200001)36:1<78::AIDDEV8>

3.0.CO;2-5

Csatádi K, Kustos K, Eiben C, Bilkó Á, Altbäcker V, 2005. Even minimal human

contact linked to nursing reduces fear responses toward humans in rabbits. Appl

Anim Behav Sci 95 (1-2): 123-128. https://doi.org/10.1016/j.applanim.2005.05.002

Thank you for your question. As all the rabbits reared and sold by Charles River Laboratories, also the rabbits included in this study were submitted to their “Rabbit Breeding Programme”, which also includes a Handling Program designed in two phases, from their birth until their departure. When arrived at your facility, we then followed Charles River preconisation, which states: «  At your facility, we recommend petting the rabbit in your arms for a minimum of two minutes, twice a day, throughout the acclimatisation period. If the rabbit is too stressed, petting can be offered directly in the cage.”

We added this information at the end of the paragraph 2.1 (lines 126-132)

2.3. Experimental design

Indicate the start and end date of the trial.

Thank you for this suggestion, we added this information at the beginning of the paragraph 2.3 (line 150).

State the weights of the males and females, as well as the differences detected throughout the study.

Thank you for this suggestion. Writing the article, we decided to exclude the data divided between males and females since no differences were observed. We added a sentence in lines 280-281.

Where was the blood sample obtained from and with what material? How were the rabbits collected? Please indicate the handling method used to prevent injuries.

Thank you for your question. Blood was sampled in the saphenous vein. The handling was performed by a professional Veterinary Specialist Assistant  which possesses also the qualification in Use and protection of laboratory animals  as Applicator, according to the European Directive 2010/63/EU and French Law 2013-118. The rabbit was gently kept with its head under the arm of the operator, to expose the back towards the vet surgeon. With the other hand, the operator gently extended the posterior leg and, at the same, with the same hand, produced a pressure all around the knee to temporally stop the circulation and facilitate the saphenous vein identification and sampling. When the vein was penetrated by the needle, the knee pressure was stopped to facilitate blood flow and aspiration into the syringe. One the sampling completed, a pressure on the zone of sampling was executed to avoid eventual bleeding and/or hematomas. If needed, a bandage was temporally applied. We have added a short recap of this in the chapter 2.3.

What parameters were analysed in the blood? In which laboratory? Is it accredited according to ISO 17025 or does it have a certified quality system?

Thank you for your question. Blood analysis was performed using an IDEXX LaserCyte machine that we have in our labs, provided and regularly checked by IDEXX, according to manufacturer instructions. The analysis included common erythrocytes values (RBC count, HCT, HGB, MCV,MCH, MCHC, RDW, retic%, retic) and leucocytes values (total WBC, lymphocytes, neutrophils, monocytes, basophiles, eosinophils - as % and absolute values for all – PLT).

Blood sample was performed only at day 5, as we do every time that we receive a new group of animals, to check their health status.  In this case, we have chosen to do this analysis five days rabbits’ arrival to avoid to make the blood sample right after their arrival, to limit to add these two stressful events, and we have thus chosen to couple with the day five session.

The analysis was not done to obtain results for the study, thus the results were just checked by the veterinary surgeon in order to assess the good health status of the rabbits, for these reasons the results were not included in the results presentation.

Indicate the reference bibliography for the analytical parameters in blood.

As for other cases in our facilities, as normal ranges we used the Chapter 3  (Clinical Biochemistry and Hematology) of the book “The Laboratory Rabbit, Guinea Pig, Hamster, and Other Rodents” (edited by: Mark A. Suckow, Karla A. Stevens and Ronald P. Wilson in 2012).

What video camera was used? Indicate brand and model

Thank you for this suggestion we added the camera used in the paragraph 2.4, line 180.

Were the rabbits previously dewormed?

We thank the reviewer for the question. Rabbits were not dewormed. Since they arrived from Charles River labs (Lyon, France), thus reared in an extremely controlled environment and sold as VAF/Plus® (SPF) also for intestinal parasites,  we did not considered that they needed to be dewormed.

  1. Results

It is important to differentiate the results between males and females or adequately justify why this was not done.

Thank you for this suggestion. Writing the article, we decided to exclude the data divided between males and females since no differences were observed. We added a sentence at the end of the paragraph 3.2.  For this reason, we think that it is not necessary to add a table with  all the parameters separated between males and females, since it will add a huge amount of data that can dilute the most important findings without adding particular knowledge.

The average weights in each of the weighings must be reflected in a table, differentiating females and males

As stated here above, data are not divided between males and females. Anyway, we added daily weight data in table 3.

The levels of the parameters measured in blood on each of the dates must be reflected in a table, differentiating females and males.

Thank you for your question. As stated here above, we did not include these results in the paper since they were not intended as  necessary for the aim of the study. The reason of this unique blood analysis was to check rabbits’ health status, after the transport and first days of acclimation in the novel facilities. As decided during the redaction of the manuscript, also now we don’t think that these data can add something to the article, and also they may just add confusion regarding the aim and the message of the work.

  1. Discussion

Indicate the advantages of using pheromones compared to medications to relieve or prevent stress.

Thank for this comment, which is very interesting and important. We have added this aspect into the discussion (lines 335-343)

Round 2

Reviewer 3 Report

Comments and Suggestions for Authors

New reviewer's note: It is important to justify why there are no differences between the sexes. There is a bibliography on this subject. This justification is based on the need to differentiate between acute and non-chronic stress. Differentiation should be incorporated into the introduction as indicated in the initial review.

Author Response

New reviewer's note: It is important to justify why there are no differences between the sexes. There is a bibliography on this subject. This justification is based on the need to differentiate between acute and non-chronic stress. Differentiation should be incorporated into the introduction as indicated in the initial review.

Thank you for your comments.

We have added in the introduction (lines 60-82) the part concerning the physiological basis of the stress, differentiating between acute and chronic stress, also adding the bibliography that you kindly suggested in the 1st review round.

We have also add, into the discussion (lines 365-372), an explanation for the absence of differences between males and females results, adding the bibliography to support this explanation.